# Accessibility to Occupational Therapy Services for Hereditary Transthyretin Amyloidosis

**DOI:** 10.3390/ijerph19084464

**Published:** 2022-04-07

**Authors:** Aina Gayà-Barroso, Juan González-Moreno, Adrián Rodríguez, Tomás Ripoll-Vera, Inés Losada-López, Margarita Gili, Eugenia Cisneros-Barroso

**Affiliations:** 1Internal Medicine Department, Son Llàtzer University Hospital, 07198 Palma de Mallorca, Spain; ainaisabel.gaya@ssib.es (A.G.-B.); jgonzalez4@hsll.es (J.G.-M.); adrian.rodriguez@hsll.es (A.R.); ialosada@hsll.es (I.L.-L.); 2Health Research Institute of the Balearic Islands (IdISBa), Son Llàtzer University Hospital, 07198 Palma de Mallorca, Spain; tripoll@hsll.es; 3Cardiology Department, Son Llàtzer University Hospital, 07198 Palma de Mallorca, Spain; 4Department of Psychology, Spain Health Research Institute of the Balearic Islands (IdISBa), University of Balearic Islands, 07122 Palma de Mallorca, Spain; mgili@uib.es

**Keywords:** transthyretin, amyloidosis, occupational therapy, polyneuropathy, accessibility

## Abstract

This study was designed to investigate the global utilization of occupational therapy (OT) services by patients with hereditary transthyretin amyloidosis (ATTRv) in Spain. The main objective was to find out whether these patients have access to OT services and the types of interventions being offered to them, together with their satisfaction and real benefits as users. We developed an online questionnaire which was distributed to patients with ATTRv in Spain through patient associations. Seventy-four patients with a diagnosis of ATTRv residing in Spain participated in the study. Thirteen had already used OT services at least once, felt that OT interventions improved their quality of life, would recommend OT services to others, and would return to see an occupational therapist. However, 61 had never used this type of service before. Of these, 35 knew what OT is and 13 declared that they considered that OT interventions in ATTRv could be positive for them. The results suggest that the use of OT services by ATTRv patients is low, mainly because of the lack of information about the occupational profile of individuals with this disease. The low response rate obtained for the survey limits generalization, and thus further research to confirm these preliminary findings is needed.

## 1. Introduction

Hereditary transthyretin amyloidosis (ATTRv) is characterized by a buildup of amyloid deposits consisting of amyloid fibrils derived from the accumulation of unstable conformations of the transthyretin (TTR) protein in different tissues [1,2,3,4,5,6,7,8,9,10]. Currently, more than 140 mutations in the *TTR* gene have been described, with Val50Met reported as the most common variant in ATTRv patients, and with polyneuropathy as the main manifestation (ATTRv-PN) [2].

ATTRv is a disease with varying presentations and symptomatology among individuals from different geographical areas [1,2,3,4,5,6,7,8]. Symptoms are mainly neuropathic in the V50M variant, but gastrointestinal disorders, cardiomyopathy, nephropathy, and/or ocular deposition can present in association with the neurological involvement [1,2,3,4,5,6,7,8,9].

ATTRv is usually characterized by a first phase with mild autonomic dysfunction (gastroparesis, constipation, diarrhea, or postural hypotension) and sensory impairment mainly in the lower limbs. In the second phase, there is a progression of the autonomic dysfunction and there may be sensory-motor impairment of the upper limbs, so that help is usually needed for daily activities. In the third phase, dysfunction is severe and even complete paralysis may develop, with death occurring approximately 10–13 years after onset of symptoms [3,4,5] if ATTRv is not treated properly with the existing treatments.

ATTRv is a disease that usually manifests in adulthood and its progressive course can extend over more than a decade. Consequently, the onset and progression of the disease can have a significant impact on patients’ lifestyles and well-being, and as a consequence, on their families. This pathology is highly limiting on a physical, mental, occupational, and social level as a result of the significant impairment on everyday life activities. Thus, occupational performance is altered by the limitations experienced by the patients in their environment [10].

The World Federation of Occupational Therapists (WFOT) defines Occupational Therapy (OT) as “A profession concerned with the promotion of health and well-being through occupation” [10]. It is therefore the role of the occupational therapist to intervene when there is an occupational risk or disturbance, whether due to physical, psychological, social, or environmental causes, to enable patients with an ATTRv diagnosis to develop, recover, and/or compensate for the necessary skills for the correct performance of meaningful activities, as independent functioning guarantees health and well-being.

Several studies have addressed the effectiveness of OT and other interventions aimed at facilitating patients’ daily living in other rare diseases such as Charcot-Marie-Tooth (CMT) neuropathy, a condition with a varying severity of disability similar to ATTRv. To date, however, no interventions have been described in the literature from an occupational perspective in patients with ATTRv even though it is likely that patients with this chronic disease would significantly benefit from the implementation of an occupational intervention. The aim of this study was to investigate general accessibility to specific occupational programs for patients with ATTRv in Spain.

## 2. Methods

### 2.1. Design

This was a descriptive, cross-sectional study designed to explore current knowledge about OT among a population of ATTRv patients divided into the following three study subgroups: asymptomatic TTR mutation carriers, patients in early stages of the disease (I, II), and patients in advanced stages (III, IV).

In addition, the degree of accessibility to this branch of health care was explored, for which an online questionnaire was created by the research team. Participants were allowed to have assistance in answering questions when needed because of the severity of their symptoms.

Ethical approval was granted by the Ethics Committee of the Balearic Islands and the Research Commission of Hospital Universitario Son Llàtzer (Decision number: IB 4587/21 PI). As the survey was anonymous, consent to participate was included in the survey itself and there was no need for a separate informed consent to be signed.

### 2.2. Participants

All participants were required to have a formal diagnosis of ATTRv or to be carriers of a pathogenic TTR mutation. They were encouraged to complete the questionnaire regardless of whether they had or had not used OT services previously. Participants were recruited through patient association websites and from the Son Llàtzer University Hospital database.

### 2.3. Data Collection

The questionnaire was available online on the website of the Balearic Andrade disease association and was mailed by this association to over 200 people with ATTRv-related mutations residing in Spain, from 12 July to 30 September 2021. Several survey reminders were sent throughout the data collection period.

### 2.4. Survey

Due to the online nature of the survey, patients provided their consent to participate in the study by confirming that they had read the patient information form and that they freely agreed to participate in this study. The online survey included 19 items. The first part of the survey was designed to determine the demographic characteristics of the cohort included. Disease stage was reported using the polyneuropathy disability (PND) score, a clinical scale used in ATTRv-PN to describe patients’ functional status. Scores range from 0, no symptoms; I, sensory disturbances with preserved walking capacity; II, impaired walking capability but no need for a stick or a crutch; IIIa, need for a stick or a crutch for walking; IIIb, 2 sticks or crutches required for walking; to IV, confinement to a wheelchair or bed (Figure 1) [11].

The second part was aimed at evaluating the subjects’ general occupational abilities. The last part examined the subjects’ global knowledge about OT, their previous use of it, and their degree of satisfaction and perception of utility if they had previously used OT services. The 19-item survey included open and multiple-choice questions. Some of the items regarding OT services received were rated on a Likert scale, with a score of 1 meaning “totally disagree” and 5 “totally agree”. The higher the score, the more the individual agreed with the item evaluated. The survey is available in Appendix A.

### 2.5. Data Analysis

We performed a descriptive analysis using counts and percentages for categorical variables, and mean and standard deviation for continuous variables after assessing for normality.

IBM Statistical Package for the Social Sciences v.23 (IBM Corporation, Armonk, NY, USA) was used throughout the data preparation and analyses.

## 3. Results

Overall, 74 subjects completed the survey, and among these 7 were asymptomatic carriers. Demographic characteristics are summarized in Table 1.

Data on symptom presentation in participants are shown in Figure 1.

The main symptoms described per study subgroup were as follows: of 44 (59.4%) patients in the early stages of the disease, 3 (4.05%) reported constipation and diarrhea, 1 (1.4%) tiredness, 2 (3%) leg weakness, 1 (1.4%) vision difficulties, and 37 (50%) pain or loss of sensation in feet or legs; all 15 (20.3%) patients in advanced stages reported pain or loss of sensation in feet or legs as the main symptom, except 1 (1.4%) who reported difficulties with fine motor skills. Seven (16%) patients did not answer the disease stage question.

Patient performance of basic activities of daily living, instrumental activities of daily living, and advanced activities of daily living are described in Figure 2.

Figure 2 represents the difficulties that patients reported in basic activities (grooming, dressing, eating, functional mobility to perform basic tasks), instrumental activities (ability to use the telephone, laundry and dressing, shopping and running errands, transportation, meal preparation, medication management, housekeeping, ability to manage finances), and advanced activities of daily living, such as hobbies, studies, or work. Patients who reported an impact of the disease on their work or study plans had either had to stop working due to pain, stress, or disability, or had required adaptation in their studying or working life due to the disease.

Activities of daily living were analyzed by patient subgroup. All seven (100%) of the asymptomatic carriers included in the study reported not having difficulties in their activities of daily living.

Of the patients who were in early stages of the disease (I/II), 43 (97.7%) reported being independent and having few or some difficulties in basic activities, whereas 14 (93%) patients in advanced stages of the disease (III/IV) reported having some difficulties, important difficulties, or much difficulty.

In relation to instrumental activities of daily living, all patients in early stages reported being independent or having only some difficulties performing these activities whereas 14 (93%) patients in advanced stages reported having some difficulties, important difficulties, or much difficulty in these activities.

Results for advanced activities of daily living were similar to the other two categories, with 37 (81%) patients in early stages reporting being independent or having few difficulties, and 13 (93%) patients in advanced stages reporting having some or important difficulties, or much difficulty.

Regarding knowledge about functions of occupational therapists, 39 (52.7%) had never heard of the profession, in particular 4 (10.2%) asymptomatic, 27 (69.2%) in early and 8 (20.5%) in advanced stages. There were 61 (82.4%) participants who had never used OT services and 13 (17.6%) who had received OT services, most of them only once. In particular, 6 (46.2%) early-stage participants and 7 (53.8%) advanced-stage participants had received such services.

Among the patients who answered that they had already used OT services, 8 (61.5%) had obtained access to these services after at least two years following diagnosis, with 38.4% in early stages of their disease and 23.1% in advanced stages of their disease. There were 3 (23.1%) who had received OT services less than two years after diagnosis, all in stage I, and the other 2 (15.4%) had had access to OT services before diagnosis due to ATTRv-unrelated causes.

Of the 13 respondents who reported having used OT services, 7 (53.8%) had received physical interventions, 3 (23%) motivational interventions, 2 (15.3%) educational interventions, and 1 participant did not specify the type of intervention received. Eight (61.5%) participants were currently using OT services, 1 (7.7%) had used them a year ago, 3 (23%) more than a year ago, and 1 (7.7%) more than three years ago.

When asked about the appropriateness of having an occupational therapist as part of a multidisciplinary team for ATTRv management, 45 (67.2%) totally agreed that it would be beneficial, 8 (11.9%) agreed, 7 (10.4%) did not know, 2 (3%) disagreed, and 5 (7.5%) totally disagreed.

A Likert scale was used to determine the level of satisfaction with OT services among the 13 patients who had previously received OT: 11 (86.7%) totally agreed, 1 (6.7%) agreed, and 1 (6.7%) did not know. Participants who were neutral or not satisfied were in early stages of the disease or asymptomatic.

## 4. Discussion

Our findings show that ATTRv affects the occupational dimension and everyday life. However, most patients had never heard about OT and were not familiar with the functions of occupational therapists. Although 17.6% of participants had received an OT intervention at some point, almost half of the participants reported that they had never heard about OT before. As previously reported, lack of knowledge about OT can lead to a delay in access to OT services [12]. Moreover, time elapsed between initial symptom progression in ATTRv patients and provision of OT services was extremely delayed despite their crucial role in the implementation of compensatory strategies. The limited existing literature on this specialty also hinders accessibility to OT. Education of health care professionals across the multidisciplinary teams that treat ATTRv patients, of patients, and of occupational therapists is essential to ensure that ATTRv patients gain access to OT and receive appropriate interventions to satisfy their specific needs in a timely manner [13]. Studies on the role of OT in diseases such as CMT highlight the need for occupational therapists to gain and maintain the highest levels of competency through continuing educational projects to ensure appropriate interventions are provided to patients [13], and this could be extrapolated to ATTRv.

Although the study cohort had a mean disease course of 10 years, most patients who were currently using occupational services had started to do so less than one year ago. This means that most of them were approached by an occupational therapist more than 10 years after diagnosis. This is a significant finding since research shows that most occupational and motivational struggles appear right after diagnosis. It is therefore important that individual occupational interventions are provided soon after diagnosis to manage symptom progression and facilitate everyday life activities despite patients living with a rare disease. The findings in the CMT study were similar, with participants receiving OT services 12 years after diagnosis. As with ATTRv, this was significant considering that progressive deterioration starts in the early stages of the disease.

Among patients who reported an impact of the disease on their work or study plans, most indicated that they had had to stop working due to pain, stress, or disability. Others had had to adapt their schedule because they were unable to handle work-related stress or disease-related fatigue. Even though one of the main functions of occupational therapists is to provide strategies and adaptations that help patients to continue performing relevant occupations of their daily life, patients do not perceive these health care professionals as a key figure due to a lack of knowledge.

The last important finding concerned patient satisfaction with OT services. When asked about the suitability of integrating an occupational therapist in the multidisciplinary teams dedicated to the diagnosis and treatment of their disease, more than half of participants totally agreed that it would be suitable. Various studies in this field have shown that participants who have received OT services can experience benefits in their daily living [13].

Our results suggest that ATTRv leads to occupational impairment. However, most patients did not have access to OT services which, as shown, could benefit them. Although OT is provided in primary care in at least 14 European countries, this is not the case in Spain. OT is still underused for patients who require this service. Several factors may explain this, such as the relatively small number of OT professionals in Spain and the unfamiliarity of patients, referrers, health care professionals, and the general public with the profession [12]. In studies evaluating reasons why patients do not have access to OT, most respondents report that their primary care doctor has never recommended it [13]. It would therefore appear that it is essential to promote educational programs targeting not only end users but also other health care professionals, including those in multidisciplinary teams, to ensure all patients may receive the most appropriate and individualized care from highly qualified OT professionals.

Furthermore, the lack of standardization in the provision of OT services together with the lack of knowledge about this specialty may negatively affect this profession as patients may perceive OT services as unnecessary. Other studies show that in rare diseases such as CMT, participants usually report that their occupational therapist is unfamiliar with the disease or is not up to date on current findings [13]. Our findings suggest individuals with ATTRv are not receiving OT interventions or do not receive them at an appropriate time, although our results are not conclusive due to the small sample and the global nature of the results. Further research is thus clearly needed, including research on the outcomes of specific OT interventions used in the ATTRv population and ATTRv awareness among professionals working in multidisciplinary teams such as rehabilitation teams.

## 5. Limitations

An important limitation of this study is that recruitment of the sample was initiated through ATTRv patient associations. Consequently, the findings can only be extrapolated to patients with this disease who are members of these associations. It would be important to conduct this research study with patients whose participation is significantly restricted and who do not have the support of specialized associations to overcome such restrictions. Furthermore, the fact that participants were recruited through associations and that data collected focused on psychosocial and occupational aspects to avoid focusing on medical aspects has surely led to the fact that no additional data that may have been of interest ―such as information on patient diagnosis and medication―were collected.

This study is a first global evaluation of availability and utilization of OT services for ATTRv patients in Spain and the findings show the need for further research on specific factors that have an influence on the impact of the disease and potential strategies to address them.

Another important limitation was the small size of the cohort. Unfortunately, there is no centralized registry for this disease in Spain, which makes it difficult to reach all patients. In addition, the survey was distributed online and many aged individuals or patients in advanced stages of the disease have difficulties accessing a computer. The use of paper questionnaires or the possibility of answering the questionnaire by phone for the population in advanced stages of the disease should be considered for future studies. Finally, as the survey was available online only for a short period of time, this likely contributed to the low response rate.

## 6. Conclusions

This study has shown that ATTRv has an impact on occupational performance. However, OT was rarely used among the ATTRv cohort, mainly due to lack of awareness. Our results seem to indicate that there are no ATTRv OT specialists and there is therefore a need for training occupational therapists in the diagnosis and treatment of ATTRv so that patients may receive appropriate and valid interventions. Further research is required to determine accessibility to OT services for ATTRv patients and to measure the effects of OT interventions.

## Figures and Tables

**Figure 1 ijerph-19-04464-f001:**
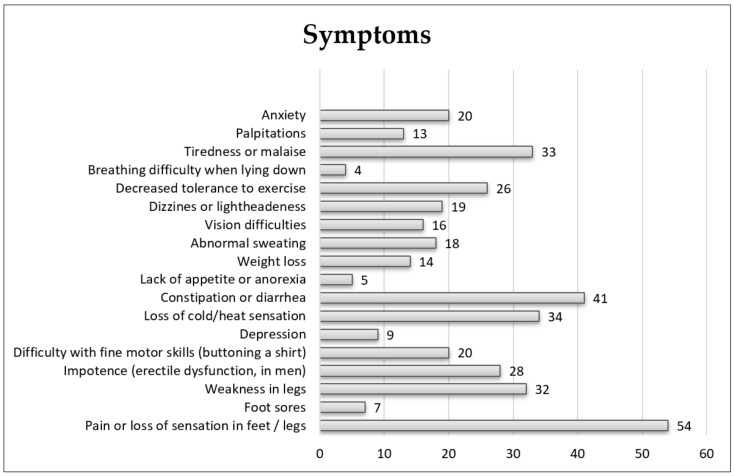
Global study population symptoms reported in the survey.

**Figure 2 ijerph-19-04464-f002:**
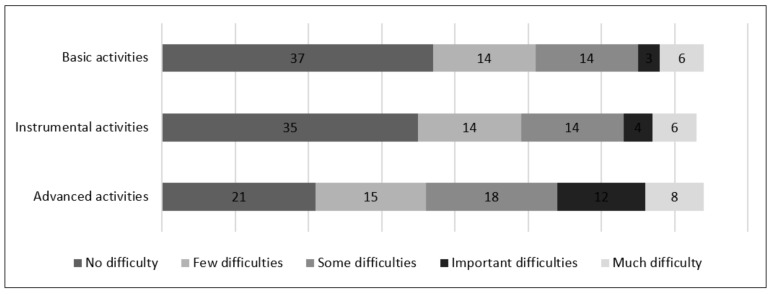
Global performance of activities of daily living.

**Table 1 ijerph-19-04464-t001:** Demographic characteristics of the study population.

Characteristics	Subjects (*n* = 74)
	Mean (±SD)
Age in years	56 (13.1)
Age at diagnosis	47 (15.6)
Age at genetic testing	46.2 (14.3)
Sex *n* (%)	
Female	28 (38.4)
Polyneuropathy disability (PND) score (%)	
0	7 (9.8)
I	29 (40.3)
II	15 (20.8)
IIIa	9 (12.5)
IIIb	4 (5.6)
IV	2 (2.8)

## Data Availability

The dataset generated during and/or analyzed during the current study is available from the corresponding author upon reasonable request.

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
