# Peer review of "Accessibility to Occupational Therapy Services for Hereditary Transthyretin Amyloidosis"

_ijerph, 2022, doi:10.3390/ijerph19084464_

Round 1

Reviewer 1 Report

Amyloidosis is a significant genetic problem in some provinces of Spain and deserves to be studied. However, this document is a descriptive manuscript mentioned by the authors. They also mention some major methodological flaws in limitations, such as sample bias and low sample size. Other flaws include bias in data collection, lack of validity of the instrument used, controls such as medication and its type. Also, the type and time of therapies received by patients and the professional who applied said therapies, besides many other methodological risk factors impossible to save for publication in this journal. Due to the previous, there is little importance described in discussion and short usefulness to IJEPRH that intends to increase its quality. I have no comments regarding the good quality of the presentation of the document, its language, order, tables, and figures.
Please check out books on bias in the scientific method.

Author Response

We appreciate the reviewer’s comments. This first study aims to approach the problem from a global point of view in patients diagnosed with ATTRv. Based on the results obtained, the next step will be to analyze the factors that influence the impact of the disease and the possible tools to address them according to the different stages of the disease. This was survey research, and as we were not able to identify any validated survey to assess the availability of OT for ATTRv patients in Spain we created our own survey. The limitations of survey research have been widely described in the literature and  have been considered in our discussion. 

Reviewer 2 Report

I applaud the authors for bringing the need for Occupational therapy in hereditary amyloidosis (infact all amyloidosis).  It is a well written paper and should be accepted after some minor edits:

Minor:

1:  on table 1:  how was the polyneuropathy disability score determined?

    Was this in the patients charts for review?

2: Line 139:  instead of “ this figure” , it may be better to say “Figure 2”

Author Response

Comments and Suggestions for Authors

I applaud the authors for bringing the need for Occupational therapy in hereditary amyloidosis (infact all amyloidosis).  It is a well written paper and should be accepted after some minor edits.

Answer: We appreciate the reviewer’s comment.

 1:  on table 1:  how was the polyneuropathy disability score determined?

    Was this in the patients charts for review?

Answer: The survey included the description of each stage as determined by the Polyneuropathy disability (PND) score. Thus, patients were able to identify the stage of their disease based on the description included in the questionnaire. The item included in the survey to record the patient’s PND score was the following (please see attachment).

2: Line 139:  instead of “this figure”, it may be better to say “Figure 2”

Answer: Line 139 has been amended to include the referee’s suggestion.

Reviewer 3 Report

The paper presents the results of an online questionnaire on the knowledge and usage of Occupational Therapy (OT) in ATTRv. Even though there are no studies to date assessing the efficacy of OT in ATTRv, a significant effect is to be expected given the positive results observed in a variety of different conditions and thus the study can have a certain relevance to the readers by raising this issue. The study design seems adequate to the task, however the analysis can be improved by increasing the detail at a subgroup level.
Important: the authors cited the questionnaire as Supplementary material but failed to upload it together with the manuscript. I would need it to complete my report.

Major concerns:

  • Introduction, lines 48-74: these paragraphs seem an excessively long presentation for occupational therapy, which is per se a well-known intervention. Please significantly shorten, possibly limiting to the literature concerning psychosocial burden in ATTRv and OT intervention (or lack of this literature).
  • Matherials and Methods: there is significant repetition/redundancy between the different paragraphs. Please check and modify accordingly
  • Results: data regarding the patients severity is presented using the Polyneuropathy Disability Score (PND), a clinical-functional scale usually rated by a clinician, however it is unclear how the authors ascertained it by an online self-compiled questionnaire.
  • Results: physical therapy and OT offer are expected to be different according to disease severity, thus I would present those results both for the whole sample and according to disease severity
  • Discussion: this chapter needs editing, aimed at comparing the results of the study to the exisiting literature in the same or similar conditions and not just repeating the results. In particular, in the absence of studies in ATTRv I would prefer a parallel with other neuropathies (10.3109/07380577.2012.755277 ; 10.1186/s13023-022-02172-5) than completely unrelated conditions as rheumatoid arthritis. I would also discuss the generalizability of these findings: the sample is enriched in early onset cases, however in most countries these are exceptional. Also, limited availability of OT services in general could be country-specific or other western countries have a similarly limited access to the service? How about other rehabilitation programs?

Minor remarks

  • lines 47-49 "Because ATTRv is a rare, hereditary, heterogeneous, progressive, and multisystemic 47 disease, patient deterioration has a strong psychosocial impact on their life and their 48 families [12]." - the reference cited deals with clinical characteristics but does not really address the impact of the psychosocial burden on the life of the patient or families. I would add a different reference, if available
  • lines 52-53 "Psychosocial profiles of families of patients with 52 other chronic diseases have already been described [13]." does not really seem necessary
  • line 125: "A p-value <0.05 was considered as statistically significant." no need as no inferential statistics is presented
  • Table I: Since ATTRv is a diagnosis of clinical/subclinical impairment in the presence of a positive genetic tests, it makes no sense for "Age at diagnosis" being lower than "Age at disease onset". I assume the authors were referring to the genetic diagnosis, thus I suggest using "Age at genetic testing" to "Age at diagnosis" to avoid misinterpretation
  • References: please double check references as they are not formatted -according to the style required by the journal. Also, ref 7-8 appear to be split.

Author Response

Answer: We appreciate the comments of the reviewer and we apologize for our mistake as we forgot to upload a copy of the survey as Supplementary material. We have now amended this mistake by uploading the survey as supplementary material. We did not perform a detailed subgroup analysis because of the small sample size included in the study. As this is the first study investigating the utilization of OT services in ATTRv patients in Spain,  we aimed to conduct a general investigation of the availability of OT for this rare disease independently of disease stage, gender, or age. However, we have also been working in the characterization of the occupational impairment caused by ATTRv disease from the identification of a subject as an asymptomatic carrier to the diagnosis of the disease, and we have found a significant impact in daily activities at all stages (unpublished data). Thus, as OT could have an impact at all the stages of the disease, we have added  in the results section a subanalysis in terms of disease stage.

Major concerns:

  • Introduction, lines 48-74: these paragraphs seem an excessively long presentation for occupational therapy, which is per se a well-known intervention. Please significantly shorten, possibly limiting to the literature concerning psychosocial burden in ATTRv and OT intervention (or lack of this literature).

Answer: A revision of the mentioned lines has been carried out and it has been adapted as suggested.

  • Materials and Methods: there is significant repetition/redundancy between the different paragraphs. Please check and modify accordingly

Answer: The materials and methods section has been rewritten as suggested, avoiding redundancy.

  • Results: data regarding the patients severity is presented using the Polyneuropathy Disability Score (PND), a clinical-functional scale usually rated by a clinician, however it is unclear how the authors ascertained it by an online self-compiled questionnaire.

Answer: The survey included the description of each stage as determined by the Polyneuropathy disability score. Patients were able to identify their stage by the description included in the questionnaire. The item included in the survey to record patient’s PND was the following (please see attachment).

  • Results: physical therapy and OT offer are expected to be different according to disease severity, thus I would present those results both for the whole sample and according to disease severity

Answer: According to the type of Occupational Therapy (OT) interventions, patients will obtain benefits depending on their symptoms and disease stage. Although this study is a global approach, the results have been described by subgroups (asymptomatic, mild, and severe patients).

  • Discussion: this chapter needs editing, aimed at comparing the results of the study to the exisiting literature in the same or similar conditions and not just repeating the results. In particular, in the absence of studies in ATTRv I would prefer a parallel with other neuropathies (10.3109/07380577.2012.755277 ; 10.1186/s13023-022-02172-5) than completely unrelated conditions as rheumatoid arthritis. I would also discuss the generalizability of these findings: the sample is enriched in early onset cases, however in most countries these are exceptional. Also, limited availability of OT services in general could be country-specific or other western countries have a similarly limited access to the service? How about other rehabilitation programs?

Answer: This comment is greatly appreciated. A revision and restructuring of the article discussion has been performed, and a comparison with a study of occupational therapy (OT) interventions in Charcot-Marie-Tooth disease (CMT), as suggested by the reviewer, has been included. The situation described in that article has been analyzed and a comparison has been made with the results obtained in our study. In addition, an analysis of the situation of the provision of OT services in other countries has been included as an important fact to take into account in this study.

Minor remarks

  • lines 47-49 "Because ATTRv is a rare, hereditary, heterogeneous, progressive, and multisystemic 47 disease, patient deterioration has a strong psychosocial impact on their life and their 48 families [12]." - the reference cited deals with clinical characteristics but does not really address the impact of the psychosocial burden on the life of the patient or families. I would add a different reference, if available

Answer: Based on this comment, the sentence has been amended and the reference has been eliminated as it is a remark based on our own observations.

  • lines 52-53 "Psychosocial profiles of families of patients with 52 other chronic diseases have already been described [13]." does not really seem necessary

Answer: Based on the reviewer’s comment, lines 52-53 have been deleted because there is no purpose of having it in the introduction.

  • line 125: "A p-value <0.05 was considered as statistically significant." no need as no inferential statistics is presented

Answer: This information has been deleted, due to the absence of inferential statics.

  • Table I: Since ATTRv is a diagnosis of clinical/subclinical impairment in the presence of a positive genetic tests, it makes no sense for "Age at diagnosis" being lower than "Age at disease onset". I assume the authors were referring to the genetic diagnosis, thus I suggest using "Age at genetic testing" to "Age at diagnosis" to avoid misinterpretation

Answer: Table I has been corrected in order to avoid misinterpretations.

  • References: please double check references as they are not formatted -according to the style required by the journal. Also, ref 7-8 appear to be split.

Answer: All references have been checked and corrected according to the style required by the journal.

Round 2

Reviewer 3 Report

The authors made a significant improvement in the quality and readability of their article.
As a last minor correction, lines 162-167 I would prefer citing the percentage of asymptomatic/early/advanced patients having knowledge of OT, to show that is not severity-dependent

i.e. "Regarding knowledge about functions of occupational therapists, 39 (52.7%) had 161 never heard of the profession, in particular 4 (50.0%) asymptomatic, 27 (50.4%) in early and 9 (50.0%) in advanced stages. Sixty-one (82.4%) participants had never used OT services and 13 (17.6%) had received OT services, most of them only once. In particular, XXX (YYY%) early stage participants and XXX (YYY%) advanced stage participants had received such services."

Author Response

We appreciate the reviewer’s comments. We have rewritten lines 162-167 according to the reviewer suggestion.